# Testing the Trust: Verification and Validation of Bayesian Segmentation under Uncertainty

**Giuseppina Carannante**[1]        CARANNANG1@ROWAN.EDU

**Nidhal C. Bouaynaya**[1]        BOUAYNAYA@ROWAN.EDU

**Dimah Dera**[2]        UVW@FOO.AC.UK

**Hassan M. Fathallah-Shaykh**[3]        HFSHAYKH@UABMC.EDU

**Ghulam Rasool**[4]        GHULAM.RASOOL@MOFFITT.ORG

[1] *Department of Electrical and Computer Engineering, Rowan University, NJ, USA*

[2] *Chester F. Carlson Center for Imaging Science, Rochester Institute of Technology, NY, USA*

[3] *Department of Neurology, University of Alabama at Birmingham School of Medicine, AL, USA*

[4] *Machine Learning Department, Moffit Cancer Center, FL, USA*

**Editors:** Accepted for publication at MIDL 2026

## Abstract

Deep learning has achieved state-of-the-art performance in medical image segmentation, yet safe clinical deployment requires rigorous verification and validation of model robustness, reliability, and uncertainty behavior. Bayesian segmentation methods are often viewed as more trustworthy because they provide uncertainty estimates that can support human decision-making, flag unreliable predictions, and mitigate risks in downstream clinical workflows. However, most prior studies evaluate these models primarily on clean test data, with limited assessment of robustness to perturbations, and without examining whether the predicted uncertainty meaningfully correlates with segmentation quality.

In this work, we conduct a comprehensive and systematic evaluation of state-of-the-art deterministic and Bayesian segmentation models across multiple datasets, corruption types, and performance metrics. Beyond accuracy-based metrics such as DSC and HD95, we analyze over- and under-segmentation trends, predictive variance, and the relationship between uncertainty and segmentation correctness. Our results show that while all models behave similarly on clean or mildly corrupted data, performance diverges significantly as perturbations increase. Models that learn and propagate uncertainty during training tend to exhibit improved robustness under severe perturbations and uncertainty estimates that better correlate with segmentation errors, suggesting potential advantages for safety-critical deployment.

**Keywords:** Image Segmentation, Trustworthiness, Uncertainty, Validation, Verification.

## 1. Introduction

Medical image segmentation is central to clinical decision-making, guiding diagnosis, treatment planning, and longitudinal monitoring across modalities such as MRI and CT. For these applications, accuracy alone is insufficient (Galati et al., 2022): clinicians must also trust how models behave when confronted with uncertainty, noise, or out-of-distribution (OOD) inputs. Bayesian segmentation models are often considered better suited for safety-critical settings because they provide uncertainty estimates that can support human–AI decision making. Yet, the majority of segmentation models remain deterministic and are evaluated primarily using accuracy-oriented metrics like the Dice Similarity Coefficient (DSC)

(Menze et al., 2014). While these measures quantify overlap, they fail to capture essential aspects of model reliability, including sensitivity to distributional shifts, resilience to noise, and the interpretability of uncertainty maps.

Uncertainty quantification has recently become a central topic in medical image analysis. While numerous studies have proposed Bayesian or ensemble-based segmentation models to estimate uncertainty (Gawlikowski et al., 2023; Goan and Fookes, 2020), most of these works stop at model development, reporting calibration scores or qualitative examples without considering the broader perspective of Verification and Validation (V&V). In other words, they often ask "How accurate is the model?" rather than "How trustworthy is it under real-world variability?".

Clinical AI deployment demands rigorous V&V: Verification: whether we are building the model right, and Validation: whether we are building the right model for real patients. Bayesian segmentation models, by explicitly modeling uncertainty, offer a principled foundation for such an analysis. Despite this increasing interest in uncertainty quantification methods, few studies have examined Bayesian segmentation models through the lens of V&V, focusing instead on developing new methods rather than systematically testing robustness and reliability.

In this work, we approach Bayesian segmentation through the lens of V&V. We systematically evaluate model robustness, uncertainty calibration, and segmentation consistency under diverse noise and perturbation types. Beyond conventional metrics, we include boundary-sensitive measures such as Hausdorff distance (HD) and over-/under-segmentation ratios, emphasizing that trustworthiness in clinical AI extends beyond Dice similarity. Our findings indicate that uncertainty-aware Bayesian approaches can exhibit more interpretable and robust behavior under distributional shifts, contributing to the broader goal of verifiable and trustworthy medical AI systems.

## 2. Related Work

### 2.1. Verification and Validation in Deep Learning

Deep learning (DL) models have automated numerous tasks in recent years; however, as their applicability increases, so do concerns regarding their trustworthiness and reliability. These issues are inherently related to the V&V process that models should undergo before deployment. An increased research interest in this area has emerged (Huang et al., 2020).

Some authors have proposed using DL itself to support the verification and validation steps, while others have released Python frameworks to automate parts of this process, from data integrity checking to model reliability assessment (Frounchi et al., 2011; Chorev et al., 2022). Although some authors have pointed out the misleading use of the term *validation* to refer only to hyperparameter tuning (Kim et al., 2020), in general, most studies focus on testing model performance under diverse conditions such as distributional shifts, adversarial perturbations, and OOD data (Christin et al., 2021; Javed et al., 2024; Hong et al., 2024).

Building on these general investigations, a subset of research has concentrated on computer vision tasks, in particular, segmentation, where robustness and generalization are especially critical for downstream decision-making. In this context, several studies have examined how segmentation performance degrades under input corruptions (e.g., noise or adversarial attacks) (Kamann and Rother, 2021), while others address the *verification* as-

pect by analyzing how architectural choices influence robustness and performance (Arnab et al., 2018). More recently, these analyses have extended to large foundation models and their sensitivity to input perturbations (Schiappa et al., 2024).

A similar focus is observed in the medical imaging domain, where segmentation accuracy often underpins diagnostic or treatment decisions. Many researchers have emphasized the need to move beyond maximizing accuracy towards evaluating robustness, generalization, and reliability (Galati et al., 2022). Most studies in medical imaging and segmentation examine robustness and generalization under adversarial attacks (Liu et al., 2021), with some exploring how model architecture impacts adversarial robustness (Paschali et al., 2018; Rodriguez et al., 2022).

Recent reviews summarize strategies for improving robustness and generalizability, highlighting key factors such as appropriate statistical analyses, cross-validation strategies, computational complexity, validation with OOD or adversarial samples, and the role of architecture, data quality, and algorithmic design (Tran et al., 2025; Javed et al., 2024). Collectively, these works underscore that achieving reliable and trustworthy AI systems requires a systematic V&V process encompassing robustness, uncertainty, and generalization.

## 2.2. Bayesian Learning

In Bayesian models, all parameters, i.e., the neural network (NN) weights $\mathcal{W}$, are treated as random variables with a prior distribution $\mathcal{W} \sim p(\mathcal{W})$. Given a training dataset $\mathcal{D} = \{(\mathbf{X}^{(i)}, \mathbf{y}^{(i)})\}_{i=1}^{n}$, Bayes' theorem allows us to infer the posterior distribution $p(\mathcal{W}|\mathcal{D})$. From this, we can derive the predictive distribution for an unseen input $\mathbf{X}^*$ as: $p(\mathbf{y}^*|\mathbf{X}^*, \mathcal{D}) = \int p(\mathbf{y}^*|\mathbf{X}^*, \mathcal{W}) \, p(\mathcal{W}|\mathcal{D}) \, d\mathcal{W}$, where $\mathbf{y}^*$ is the associated output. The predictive distribution encapsulates all the information about the model output. Its mean corresponds to the network's prediction, while its predictive variance quantifies the model's uncertainty in that prediction. However, performing exact Bayesian inference in NN is computationally intractable due to the non-linearity of NN and the high dimensionality of the parameter space (Blundell et al., 2015). This has motivated the development of approximation methods for scalable Bayesian learning (Gawlikowski et al., 2023; Goan and Fookes, 2020).

One of the most widely used approximation techniques for Bayesian learning is Variational Inference (VI), which formulates Bayesian inference as an optimization problem (Graves, 2011). In VI, a tractable variational distribution $q_\theta(\mathcal{W})$ is introduced to approximate the true posterior $p(\mathcal{W}|\mathcal{D})$. The optimal parameters $\theta^*$ are obtained by minimizing the Kullback–Leibler (KL) divergence between the two distributions, which leads to the Evidence Lower Bound (ELBO) objective: $\mathcal{L}(\theta) = -\mathbb{E}_{q_\theta(\mathcal{W})}[\log p(\mathcal{D}|\mathcal{W})] + \mathrm{KL}(q_\theta(\mathcal{W}) \, \| \, p(\mathcal{W}))$. Depending on the choice of prior, the variational family, and the strategy used to approximate the expectations in the ELBO, several practical Bayesian DL formulations have been proposed. For instance, Gal and Ghahramani (2016) demonstrated that applying dropout during training and inference can be interpreted as performing VI. Alternatively, approaches under the umbrella of Variational Density Propagation (VDP) directly propagate both first and second moments through network layers, maintaining explicit representations of mean and covariance (Dera et al., 2021; Carannante et al., 2024). These moment-propagation schemes offer a principled and computationally efficient way to learn predictive uncertainty jointly with the network parameters.

In parallel, non-Bayesian strategies have been explored to capture uncertainty information, such as test-time data augmentation and deep ensembles (Lakshminarayanan et al., 2017; Wang et al., 2019). These approaches, while not grounded in Bayesian theory, have shown competitive empirical performance and are easier to implement in practice. Bayesian and non-Bayesian models have been compared in terms of the quality of their uncertainty estimates, with several studies examining performance under distributional shifts (Ng, 2020) and others relating uncertainty magnitude to prediction correctness (Scalco et al., 2024). Similar ideas have been used to assess prediction quality in the absence of ground truth (Sikha et al., 2025), and to detect distributional shifts or domain changes (Soufi et al., 2025; Ovadia et al., 2019; Carannante et al., 2022).

### 2.3. Uncertainty Estimation in Segmentation Models

In the context of semantic segmentation, most research has focused on practical approximation techniques. Two of the most widely adopted approaches are Monte Carlo (MC) dropout and model ensembles, which are favored for their simplicity and compatibility with existing architectures (Kendall et al., 2015; Kamnitsas et al., 2017; Ghoshal et al., 2021). In MC-Dropout, the well-known dropout regularization technique is applied not only during training but also at inference time. Multiple forward passes are performed through the network, and uncertainty is quantified using the variance or entropy of the resulting predictions. Similarly, ensemble-based methods train multiple independent (deterministic) networks; during inference, the same input is fed through all models, and the variability across their predictions provides an estimate of the uncertainty.

Recently, we proposed the SUPER-Net approach for segmentation, where the posterior distribution is approximated by its first two moments (mean and covariance) and propagated through the network layers during training (Carannante et al., 2025). At nonlinear layers, the transformation of the mean and covariance is approximated using a first-order Taylor series expansion, enabling the network to jointly learn both predictions and uncertainty in a principled and computationally efficient manner.

While Bayesian models offer a principled framework for uncertainty estimation for segmentation, prior works have typically evaluated them under a limited range of perturbations or datasets. In this work, we frame Bayesian segmentation within the context of *Verification and Validation*, assessing robustness, uncertainty sensitivity, and the correspondence between prediction errors and uncertainty across diverse distributional shifts.

## 3. Methodology: V&V of Bayesian Segmention Models

Our experiments are structured within a V&V framework for medical image segmentation models. To this end, we evaluate deterministic and Bayesian models on clean test data, study performance under multiple distributional shifts, including several noise types and adversarial attacks, and analyze robustness using complementary metrics.

A key component of our validation is the behavior of uncertainty estimates. We evaluate how predictive uncertainty changes under increasing noise levels and whether uncertainty appropriately reflects prediction errors (i.e., higher uncertainty for incorrect or unstable segmentation). This allows us to assess not only robustness but also the trustworthiness of the uncertainty information provided by Bayesian and approximate-Bayesian models.

Table 1: Model architecture and training details for each dataset.

| Dataset | Encoder filters | Decoder filters | Epochs | Batch size |
|---|---|---|---|---|
| Lungs | 16, 32, 64 | 32, 16 | 50 | 10 |
| Hippocampus | 32, 64, 128 | 64, 32 | 100 | 20 |
| BraTS | 64, 128, 256, 512, 1024 | 512, 256, 128, 64 | 100 | 20 |

### 3.1. Datasets

We employ three publicly available medical image segmentation benchmarks: Lung CT, Hippocampus MRI, and Brain Tumor MRI (BraTS) (Ma et al., 2020; Antonelli et al., 2022; Menze et al., 2014). We report results for clinical data in Appendix C. For all datasets, we use an 80/20 split for training and validation at the slice level, applied consistently across experiments. Preprocessing includes intensity normalization, removal of empty slices, and resizing or cropping to a fixed spatial resolution.

The Lung dataset consists of 20 chest CT scans with annotations for the left and right lungs, as well as infection regions (Ma et al., 2020). Binary segmentation is performed (lung vs. background). The Hippocampus dataset includes 394 MRI scans from the Medical Segmentation Decathlon (Antonelli et al., 2022), with labels for anterior and posterior hippocampus regions. The BraTS dataset comprises multi-modal MRIs from high-grade glioma patients (Menze et al., 2014). Each case includes five tissue classes; evaluation follows the BraTS convention (whole tumor, core, and enhancing regions).

### 3.2. Models Specifics

We adopt the U-Net architecture as the backbone for all segmentation experiments (Ronneberger et al., 2015). U-Net is an encoder–decoder convolutional NN with skip connections between symmetric layers, enabling the combination of high-resolution spatial information from the encoder with features from the decoder. Each convolutional block consists of two convolutional layers (kernel size of $3 \times 3$) followed by batch normalization and ReLU activation, with max-pooling used for downsampling and upsampling in the decoder. The number of filters in each encoder and decoder stage for each dataset is summarized in Table 1.

As a baseline, we train a deterministic U-Net. We then compare it against three uncertainty-aware variants: (i) MC-Dropout, using 20 MC samples with a dropout probability of $p = 0.5$ applied at the bottleneck layers as in (Kendall et al., 2015); (ii) Ensemble, comprising five independently initialized U-Nets whose predictions are aggregated at inference; and (iii) SUPER-Net, which jointly learns prediction and uncertainty by propagating mean and covariance through all layers as in (Carannante et al., 2025). All models are trained using the specifications listed in Table 1. We employ the Adam optimizer with a learning rate of 0.001, and apply early stopping based on validation performance.

### 3.3. Distributional Shifts

To evaluate robustness and generalizability, we introduce controlled distributional shifts at test time by corrupting the images with noise and adversarial perturbations. Medical images are known to be affected by acquisition and reconstruction noise, which is commonly modeled using additive white Gaussian noise, though other noise types such as Poisson,

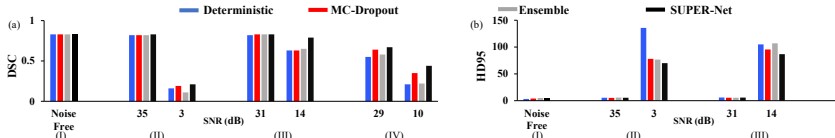

Figure 1: Performance comparison on the Lungs dataset under noise-free conditions (I), Gaussian noise applied to the whole image (II) or lung pixels only (III), and untargeted adversarial attacks (IV). HD95 and DSC are shown on the $y$-axis of (a) and (b), with SNR values on the $x$-axis for noisy conditions.

Speckle, and Salt-and-Pepper have also been reported in the literature (Goyal et al., 2018). Accordingly, we generate noisy test sets by adding Gaussian noise at multiple Signal-to-Noise Ratio (SNR) levels, and we additionally include Poisson, Speckle, and Salt-and-Pepper noise for completeness. Noise is applied either to entire image scans or selectively to the structure of interest to simulate localized corruption.

To examine vulnerability to adversarial perturbations, we apply the Fast Gradient Sign Method (FGSM) for untargeted (Liu et al., 2017) and Projected Gradient Descent (PGD) for targeted variants (Madry et al., 2018). For PGD, we use a step size of 1 and a maximum of 20 iterations. For each targeted attack, we select a source class and a target class. The attack aims to misclassify pixels belonging to the source class as the target class. For all attacks, the perturbation magnitude is controlled by the parameter $\epsilon$, which we vary to generate attacks of increasing strength. To allow a unified comparison across noise and adversarial perturbations, attack strength is reported in terms of SNR.

The selected SNR ranges span from mild to severe corruption to reflect both realistic acquisition variability and stress-test conditions. Higher SNR values correspond to common clinical variability, such as scanner noise, reconstruction artifacts, and protocol-dependent differences, while lower SNR values simulate rare but critical failure scenarios that may arise from extreme acquisition artifacts or intentionally perturbed inputs (Finlayson et al., 2019).

### 3.4. Evaluation Metrics

To assess segmentation performance, we use both region-based and boundary-based metrics. The DSC measures voxel-wise overlap between the prediction and ground truth, capturing overall spatial agreement. Complementarily, the HD quantifies the largest boundary deviation between two segmentations. Because the maximum HD is highly sensitive to isolated outliers, often caused by noise or small contour errors, we report the 95th-percentile Hausdorff Distance (HD95), a robust variant commonly used in medical imaging evaluation. Beyond these standard accuracy metrics, we evaluate over-segmentation ($O_s$) and under-segmentation ($U_s$) rates (Mou et al., 2021). These quantify the fraction of incorrectly added or missed voxels, respectively, with both values ranging from 0 to 1, where lower values indicate better performance. Such measures are clinically meaningful, as segmentation errors have distinct consequences depending on their direction. For example, under-segmenting a tumor in radiotherapy planning may lead to undertreatment and increased recurrence

Table 2: Performance Comparison for noise-free Lungs Test Dataset

|  | Deterministic | MC-Dropout | Ensemble | SUPER-Net |
|---|---|---|---|---|
| DSC | **.83** | **.83** | **.83** | **.83** |
| HD95 | **3.17** | 4.04 | 5.1 | 4.94 |
| $O_s$ | .16 | **.15** | .17 | **.15** |
| $U_s$ | .04 | **.03** | **.03** | **.03** |

Table 3: Performance Comparison for the noise-free Hippocampus Test Dataset

|  | Anterior | | | | Posterior | | | |
|---|---|---|---|---|---|---|---|---|
|  | Deterministic | MC-Dropout | Ensemble | SUPER-Net | Deterministic | MC-Dropout | Ensemble | SUPER-Net |
| DSC | **.79** | **.79** | **.79** | **.79** | .76 | .76 | **.77** | .74 |
| HD95 | **1.56** | 1.68 | 1.58 | 1.62 | **1.81** | 2.11 | 2.01 | 2.21 |
| $O_s$ | .15 | .13 | **.09** | .15 | .17 | .15 | **.11** | .14 |
| $U_s$ | **.14** | .17 | .19 | .15 | **.15** | .19 | .21 | .23 |

Table 4: Performance Comparison for noise-free BraTS Test Dataset

|  | Whole | | | | Core | | | | Enhancing | | | |
|---|---|---|---|---|---|---|---|---|---|---|---|---|
|  | Det | MC-Drop | Ensemble | SUPER-Net | Det | MC-Drop | Ensemble | SUPER-Net | Det | MC-Drop | Ensemble | SUPER-Net |
| DSC | .77 | .77 | .76 | **.83** | .58 | .58 | .60 | **.64** | .57 | .57 | .63 | **.69** |
| HD95 | 7.18 | 6.83 | 6.38 | **3.38** | 6.95 | 6.40 | 5.53 | **4.36** | 5.94 | 5.91 | 4.09 | **3.13** |
| $O_s$ | .11 | .11 | **.06** | .10 | .29 | .24 | .14 | **.08** | .41 | .40 | .29 | **.25** |
| $U_s$ | .22 | .21 | .27 | **.14** | **.21** | .26 | .34 | .34 | **.11** | .12 | .16 | .15 |

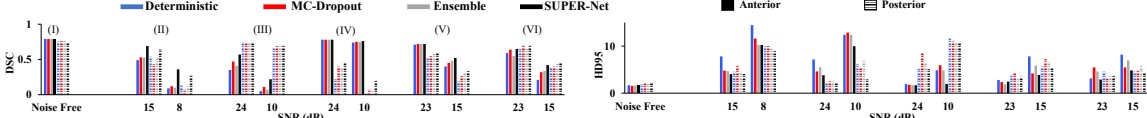

Figure 2: Performance comparison on the Hippocampus dataset for noise-free data (I), Gaussian noise applied to the whole image (II), anterior pixels (III), or posterior pixels (IV), and targeted adversarial attacks (V–VI) with source→target label mappings: 1→2, and 2→1, respectively. HD95 and DSC are shown on the $y$-axis, with SNR on the $x$-axis for noisy cases. Full-color bars correspond to the anterior structure, and dashed bars to the posterior structure.

risk, whereas over-segmenting may unnecessarily expose healthy tissues to radiation. Likewise, systematic $O_s$ or $U_s$ in longitudinal tumor monitoring can lead to misinterpretation of growth or stability.

## 4. Results and Discussion

We emphasize that the goal of this section is not to establish a single superior segmentation model, but to analyze how different uncertainty estimation strategies behave under V&V stress tests. Performance differences should therefore be interpreted in the context of robustness trends, uncertainty behavior, and failure modes rather than isolated metric improvements.

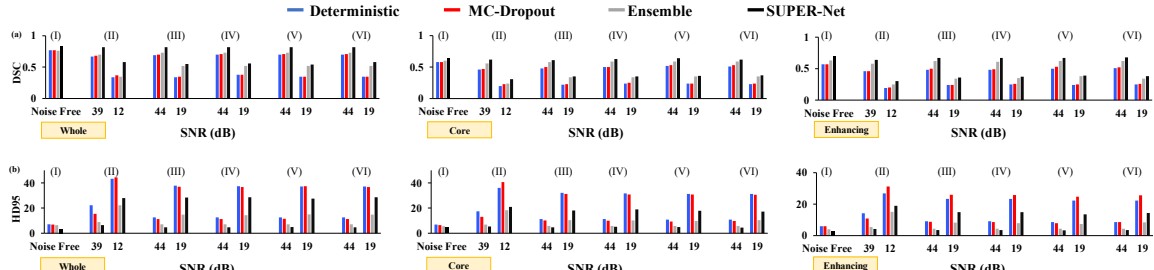

Figure 3: Performance comparison on BraTS for (I) noise-free, (II) untargeted, and (III–VI) targeted attacks with source→target label mappings: 3→1, 1→3, 3→2, and 2→3, respectively. Subplots show DSC and HD95 vs. SNR for whole tumor, core, and enhancing regions.

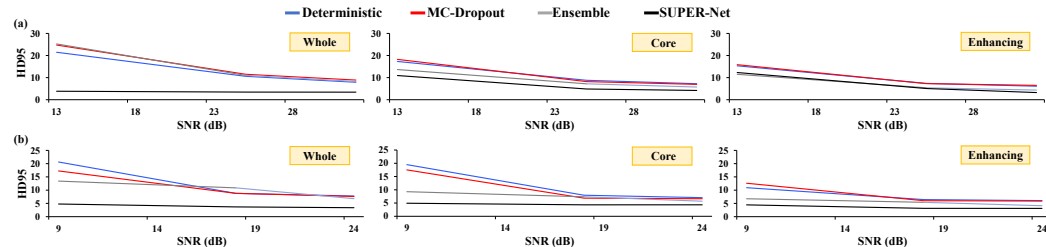

Figure 4: Performance comparison under Gaussian noise applied to (a) tumor pixels only and (b) entire scans in the BraTS test data. Subplots show HD95 across SNR levels for whole tumor, core, and enhancing regions.

## 4.1. Performance Comparison

To establish a baseline, we first evaluate all models, Deterministic, MC-Dropout, Ensemble, and SUPER-Net, on noise-free test data. Tables 2, 3, and 4 report DSC, HD95, $O_s$, and $U_s$ for the Lung, Hippocampus, and BraTS datasets. No single method consistently achieves the best performance across all metrics or anatomical structures, underscoring the importance of multi-metric evaluation.

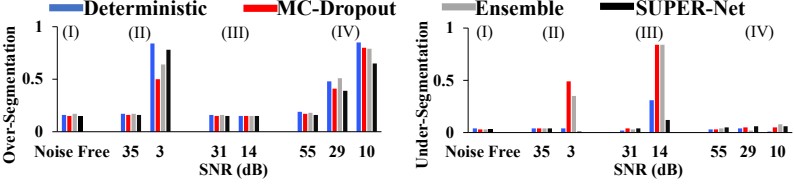

Figure 5: Performance comparison on the Lungs dataset for noise-free (I), Gaussian noise applied to the whole image (II) or lung pixels only (III), and untargeted adversarial attacks (IV). Subplots show $O_s$ and $U_s$ across SNR levels.

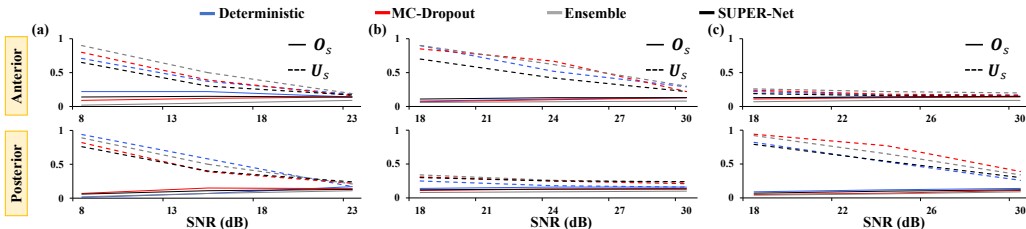

Figure 6: Performance comparison on the Hippocampus dataset with Gaussian noise applied to the whole image (I), anterior pixels (II), or posterior pixels (III). $O_s$ and $U_s$ are plotted versus SNR for the anterior and posterior structures.

Figures 1, 2, and 3 summarize segmentation performance under several perturbation scenarios, including multiple levels of Gaussian noise (applied either to entire images or only to target structures) and both targeted and untargeted adversarial attacks. Across datasets, all models maintain comparable accuracy under mild perturbations, but performance deteriorates substantially as noise or adversarial strength increases, particularly for HD95, which is highly sensitive to boundary distortions. For BraTS, the robustness trends are further illustrated in Fig. 4, which displays HD95 for Gaussian noise applied either globally or to tumor regions only.

In general, DSC and HD95 provide consistent signals of degradation, although the relative ordering of models may differ at high corruption levels. Across datasets, ensemble models tend to retain higher DSC than MC-Dropout at lower SNR levels, particularly under strong noise and adversarial perturbations. This suggests that model diversity in ensembles provides additional robustness benefits that are not fully captured by stochastic MC regularization alone. SUPER-Net exhibits the smallest increase in HD95 as noise intensity grows, suggesting comparatively greater boundary stability under these perturbations.

We additionally examine $O_s$ and $U_s$ behavior under both Gaussian noise and adversarial attacks (Figs. 5, 6, 8, 7, 9). No approach dominates across all conditions, highlighting the intrinsic tradeoff between $O_s$ and $U_s$. Some models systematically under-segment, while others shift between under- and over-segmentation depending on dataset and corruption level. However, we observe that low $O_s$ values at high noise levels may correspond to severe under-segmentation, emphasizing the need to interpret these metrics jointly with DSC and visual inspection. Across several scenarios, SUPER-Net exhibits a competitive balance across metrics and frequently ranks among the lowest combined segmentation errors, though no single method dominates across all conditions.

In addition, we provide a $k-$fold cross-validation analysis for SUPER-Net in Appendix A. Results are consistent across folds, indicating robustness is not due to fold-specific effects. We further assessed the robustness differences using statistical significance testing and 95% confidence intervals; detailed results and tests are reported in Appendix B.

### 4.2. Noise Analysis

We extend the noise analysis to additional corruption types commonly observed in medical imaging, Speckle, Salt & Pepper, and Poisson, to further assess model behavior under real-

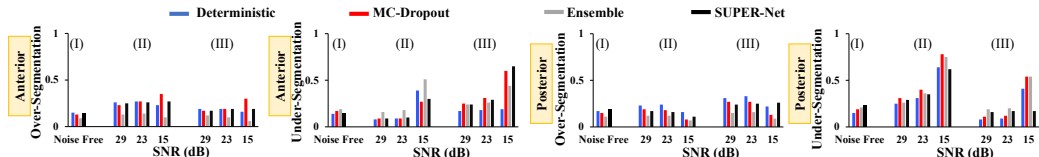

Figure 7: Performance comparison on the Hippocampus dataset for (I) noise-free, and (II–III) adversarial attacks. Panels (II) and (III) show targeted attacks with source→target label mappings: 1→2, and 2→1, respectively. Subplots report $O_s$ and $U_s$ versus SNR for the anterior and posterior structures.

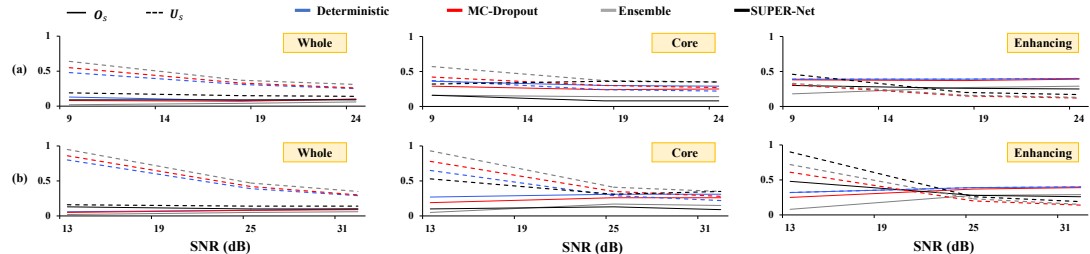

Figure 8: Performance comparison on BraTS under Gaussian noise applied to (a) the entire image and (b) tumor pixels only. The $y$-axis shows $O_s$ and $U_s$ versus SNR for whole tumor, core, and enhancing regions.

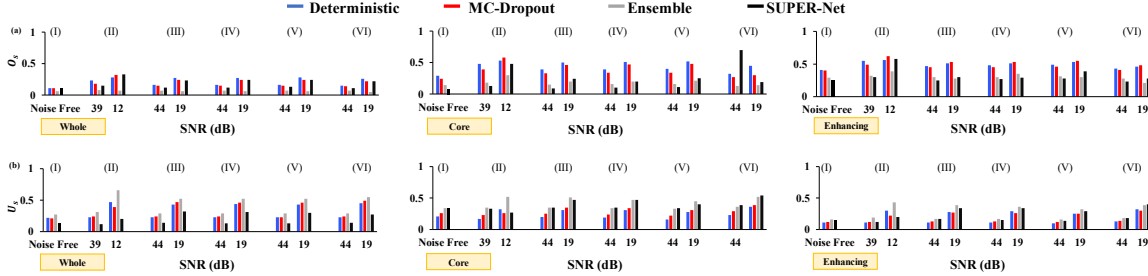

Figure 9: Performance of the four networks on BraTS for (I) noise-free, (II) untargeted, and (III–VI) targeted attacks with source→target label mappings: 3→1, 1→3, 3→2, and 2→3, respectively. Subplots show $O_s$ and $U_s$ vs. SNR for whole tumor, core, and enhancing regions.

Table 5: DSC for Hippocampus data corrupted with Speckle and Poisson noise.

| | Anterior | | | | Posterior | | | |
|---|---|---|---|---|---|---|---|---|
| | Deterministic | MC-Dropout | Ensemble | SUPER-Net | Deterministic | MC-Dropout | Ensemble | SUPER-Net |
| Speckle noise added to entire image | | | | | | | | |
| SNR ≈ 20 dB | .77 | .77 | **.78** | **.78** | .73 | .74 | **.75** | .73 |
| SNR ≈ 14 dB | .58 | .65 | .65 | **.68** | .53 | .54 | .60 | **.63** |
| SNR ≈ 10 dB | .18 | .20 | .23 | **.48** | .09 | .10 | .12 | **.37** |
| Poisson noise added to entire image | | | | | | | | |
| SNR ≈ 20 dB | .74 | .75 | .75 | **.77** | **.73** | .71 | .72 | .71 |
| SNR ≈ 11 dB | .39 | .38 | .39 | **.56** | .39 | .38 | .36 | **.52** |
| SNR ≈ 8 dB | .22 | .20 | .22 | **.40** | .23 | .21 | .21 | **.36** |

Figure 10: Performance comparison under various levels of Speckle noise applied to the anterior and posterior hippocampus. Subplots display DSC across SNR levels.

istic acquisition conditions. Performance under these noise types is reported in Table 5 and Figures 10, 11, and 12. Across all datasets and noise types, the models behave similarly under clean and low-noise conditions. However, as corruption severity increases, performance degrades sharply, especially for deterministic and approximate Bayesian methods. SUPER-Net exhibits greater robustness at low SNRs, maintaining higher DSC and more stable degradation trends, suggesting that propagating uncertainty helps increase robustness under heavy corruption. Although MC-Dropout provides uncertainty estimates, its robustness to severe perturbations is frequently comparable to that of the deterministic baseline.

### 4.3. Uncertainty Analysis

We first examine whether uncertainty responds meaningfully to corruption severity. Figure 13 shows DSC and predictive variance for SUPER-Net under Gaussian noise and targeted adversarial attacks. As expected, uncertainty increases as DSC decreases, indicating that the model expresses lower confidence when its predictions deteriorate. We then compare uncertainty behavior across all approaches. In Figure 14, we report the average predictive variance for correctly and incorrectly classified pixels under Gaussian, Poisson, and adversarial perturbations. All methods assign higher uncertainty to incorrect predictions, a desirable property for reliability under distributional shifts. Among the evaluated approaches, SUPER-Net shows a more consistently monotonic increase in uncertainty with corruption severity, whereas other methods exhibit less stable trends.

Finally, we assess whether uncertainty meaningfully identifies erroneous pixels. In Figure 15, we report the change in DSC when "uncertain" pixels are removed. Ideally, models should be uncertain only for incorrectly segmented pixels, leading to improved DSC after removal ($\Delta$DSC > 0). However, several approaches show a DSC decrease, suggesting that they also flag correctly classified pixels as uncertain. SUPER-Net shows the most consistent positive or stable $\Delta$DSC, indicating more informative uncertainty estimates.

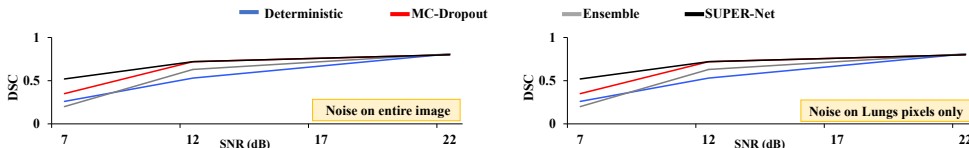

Figure 11: Performance comparison under various levels of Salt & Pepper noise applied to the entire image or the lung pixels only. DSC vs. SNR is plotted.

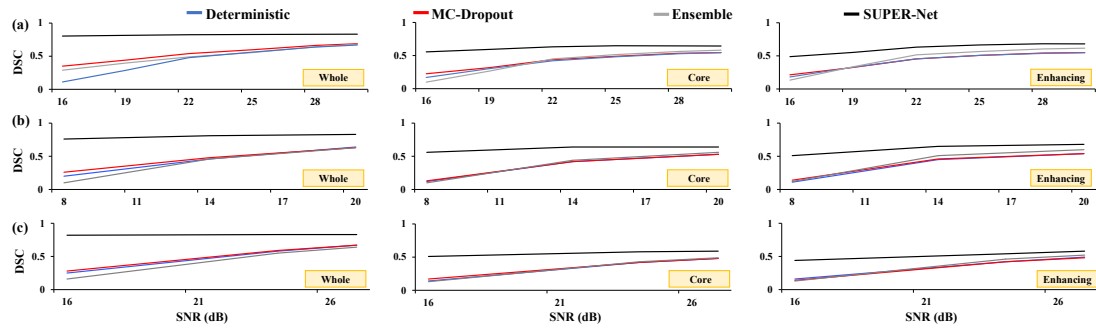

Figure 12: Performance comparison under Speckle noise added to (a) tumor pixels and (b) all pixels, and under (c) Salt & Pepper noise on the BraTS test data. Subplots show DSC across SNR levels for whole tumor, core, and enhancing regions.

## 4.4. Clinical Implications and Deployment Considerations

In clinical workflows, segmentation models are typically used as decision-support tools rather than fully autonomous systems. In this context, uncertainty-aware models offer a significant advantage by explicitly communicating prediction reliability to clinicians. Our results show that high predictive uncertainty consistently aligns with incorrect segmentations, providing a natural mechanism for directing clinician attention to regions requiring review or correction.

Such uncertainty maps could be seamlessly integrated into existing radiology or radiation therapy planning systems as visual overlays, enabling efficient physician-in-the-loop

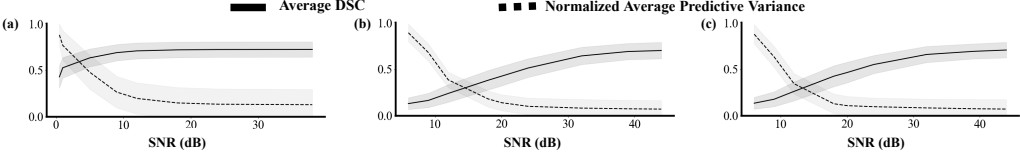

Figure 13: SUPER-Net DSC and predictive variance vs. SNR for the Hippocampus data. Variance is normalized between 0 and 1. Results are shown for (a) Gaussian noise, and targeted adversarial attacks with source→target label mappings: 1→2 (b), and 2→1 (c).

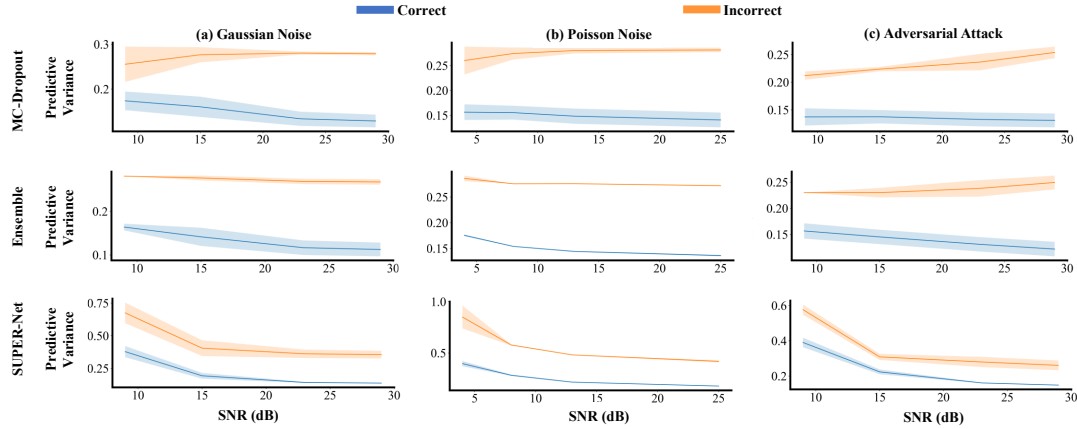

Figure 14: Average predictive variance versus SNR for: 1) correctly labeled pixels (blue) and 2) misclassified ones (orange) for (a) Gaussian noise, (b) Poisson noise and (c) untargeted adversarial attacks applied to the Hippocampus data.

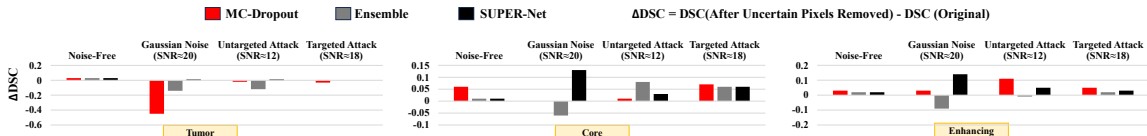

Figure 15: Change in DSC (ΔDSC) after the removal of *uncertain* pixels.

verification. By highlighting unreliable regions, uncertainty-aware models reduce the cognitive burden associated with reviewing entire segmentation outputs and help mitigate the risk of overconfident errors. These properties are particularly important in safety-critical applications such as tumor delineation, where segmentation errors may directly influence diagnosis or treatment decisions.

From a clinical perspective, these findings are particularly relevant in scenarios involving distributional shifts or malicious perturbations. For instance, diagnostic systems may encounter novel tumor appearances due to changes in imaging protocols or adversarial manipulations. In such cases, uncertainty-aware models that associate high uncertainty with unreliable predictions can help prevent overconfident errors by alerting clinicians to regions requiring closer inspection.

### 4.5. Computational Complexity Comparison

We compare the inference-time cost and memory footprint of the evaluated uncertainty estimation approaches. Table 6 reports the relative storage requirements and the average inference time (s) per slice. While deterministic models require a single forward pass, uncertainty-aware methods introduce additional computational or memory overhead depending on the approach.

The SUPER-Net framework propagates uncertainty through the network by learning an additional variance parameter per convolutional kernel. This results in a marginal increase in model parameters, but uncertainty is obtained in a single forward pass at inference time. In contrast, MC Dropout does not increase the number of learned parameters, but requires multiple stochastic forward passes to estimate predictive uncertainty, leading to increased inference time. Ensemble-based methods incur the highest memory cost, as they require storing and evaluating multiple independently trained models.

For MC-Dropout, we use 20 stochastic forward passes to estimate predictive uncertainty, following standard practice in medical image segmentation (Kendall et al., 2015). For ensemble-based methods, we evaluate an ensemble of 5 independently trained models, which prior studies have shown provides a good balance between predictive performance and computational cost (Lakshminarayanan et al., 2017). These choices ensure meaningful uncertainty estimates while keeping inference-time overhead manageable.

In clinical settings, segmentation models must not only be accurate and reliable but also computationally efficient to enable timely decision-making. From a clinical deployment perspective, inference-time and memory footprint are critical, particularly in physician-in-the-loop workflows and high-throughput settings. Sampling-based uncertainty methods incur inference costs that scale linearly with the number of stochastic passes or ensemble members. SUPER-Net incurs a modest increase in inference time due to uncertainty propagation but avoids repeated forward passes or multiple model storage.

### 5. Conclusion

Accurate and reliable segmentation is essential for many clinical tasks, including diagnosis, treatment planning, and long-term disease monitoring. Yet in practice, medical images are affected by noise, artifacts, scanner variability, and unexpected data shifts, all of which can cause segmentation models to fail silently. Reliable medical image segmentation therefore

Table 6: Comparison of memory characteristics and inference time.

| Method | # Models | Extra Params | Forward Passes | Inference Time |
|--------|----------|--------------|----------------|----------------|
| Deterministic | 1 | None | 1 | 0.81 |
| MC-Dropout | 1 | None | $N$ | 0.82 $\times N$ |
| Ensemble | $N$ | $N\times$ model size | $N$ | 0.81 $\times N$ |
| SUPER-Net | 1 | $+1$ per kernel | 1 | 1.92 |

$^{*}N$ denotes the number of MC samples or ensemble networks.

requires more than high accuracy on clean test data, it demands models that remain trustworthy under real-world variability and provide uncertainty estimates that meaningfully reflect prediction reliability for clinical use.

Through a comprehensive V&V framework, we compared deterministic, approximate Bayesian, and uncertainty-propagating segmentation models across multiple datasets and clinically relevant perturbations. Our analysis shows that approaches producing uncertainty as part of the forward pass, such as SUPER-Net, offer more stable performance under noise and adversarial conditions and generate uncertainty values that consistently flag incorrect predictions. Such behavior is crucial in clinical decision-making, where overconfident errors may directly contribute to misdiagnosis or suboptimal treatment.

Our findings underscore that model selection for clinical deployment must integrate robustness analysis, boundary-sensitive metrics, and principled uncertainty evaluation, rather than relying solely on accuracy. By framing segmentation assessment within a V&V perspective, this work emphasizes that uncertainty-aware modeling is central to building safe, interpretable, and clinically actionable AI systems. Evaluating models under distributional shifts and scrutinizing their uncertainty behavior should become standard practice in the development of trustworthy clinical AI. Ultimately, this work moves toward AI tools that better support clinicians by providing not only accurate predictions but also clear indications of when those predictions can be trusted.

## Acknowledgments

This work was supported by the National Science Foundation awards NSF 1903466, NSF 2008690, NSF 2234468, and NSF 2542166.

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

Table 7: Cross-validation results for noise-free BraTS test data

|  | Whole | Core | Enhancing |
|---|---|---|---|
| **DSC (mean ± std)** | $0.82 \pm 0.02$ | $0.66 \pm 0.02$ | $0.69 \pm 0.02$ |
| **HD95 (mean ± std)** | $3.42 \pm 0.36$ | $3.82 \pm 0.35$ | $2.93 \pm 0.34$ |

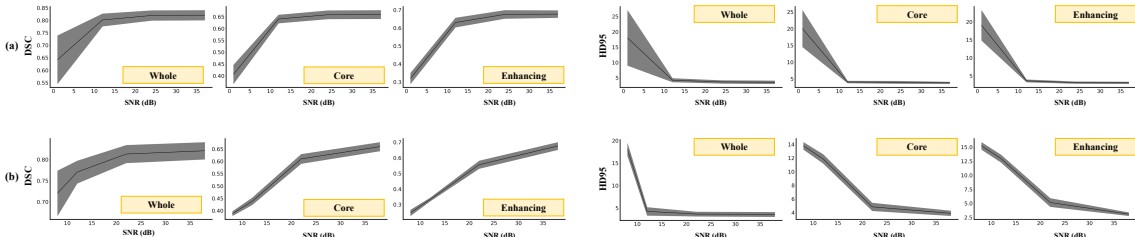

Figure 16: DSC and HD95 for SUPER-Net for Gaussian noise applied to (a) the entire image and (b) tumor pixels only of the BraTS dataset. Curves show the mean across $k = 5$ cross-validation models, with shaded regions indicating standard deviation, providing insight into the variability of the model performance.

## Appendix A. Cross-Validation

To comprehensively evaluate the performance of SUPER-Net for medical image segmentation, we conduct a cross-validation study. In our study, we opt for the $k-$fold cross-validation method with $k = 5$. Cross-validation mitigates potential biases in training and test set selection by randomly sampling different splits of the data for each iteration.

For both the Lung and Hippocampus data, the model presented above falls within the range obtained with the cross-validation results. For the Lung data, the mean DSC for the $k$-fold models is 0.83 with a standard deviation of 0.01. For the Hippocampus data, the mean and standard deviation DSC are 0.78 and 0.02 for the anterior structure, and 0.74 and 0.01 for the posterior structure.

We report the cross-validation results for the BraTS data in Table 7, which are similar to the one-split results reported in Section 4.1. Additionally, we test the reliability of the results when the $k$-fold models are tested under noisy conditions. In Figure 16 we show the DSC and HD95 vs. SNR for the $k$-fold models. The line represents the average performance, while the shaded area refers to the standard deviation.

We observe that there is not much variation among the models under low-noise conditions, while there is higher variation under high-noise conditions. Yet, it is interesting to observe that all the SUPER U-Net models from the $k$-fold cross-validation perform better than other approaches.

## Appendix B. Statistical Analysis

To support claims of robustness differences, we performed statistical significance analysis on segmentation accuracy under representative noise and adversarial conditions. Specifically, we compute 95% confidence intervals (CIs) for DSC scores using non-parametric bootstrap

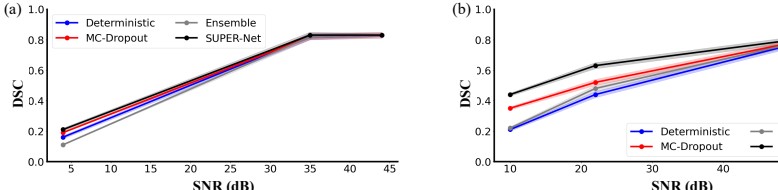

Figure 17: Mean DSCs are shown with shaded 95% confidence intervals for (a) Gaussian noise and (b) untargeted adversarial attacks applied to the Lungs test data.

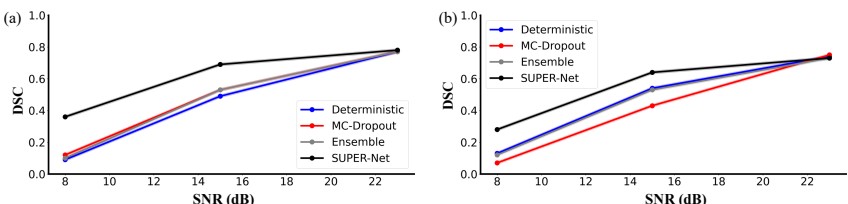

Figure 18: Mean DSCs are shown with shaded 95% confidence intervals for (a) Anterior and (b) Posterior Hippocampus for various levels of Gaussian noise applied to the whole image of the Hippocampus test data.

resampling with 1,000 iterations. Across datasets, CI half-widths were typically in the range $0.01 - 0.02$, with the wider intervals observed for the Lungs. Figures 17, 18, 19, 20, and 21 show the mean DSCs for all models across different noise and adversarial attack scenarios, with shaded areas representing the 95% confidence intervals.

In addition, we conducted paired t-tests comparing SUPER-Net against Deterministic, MC-Dropout, and Ensemble under noise and adversarial attack conditions, when a potential lack of statistical significance was observed. The results indicate that SUPER-Net achieves statistically significant improvements in most tested conditions ($\alpha = 0.05$ ). The only exception is the Core structure under an adversarial attack of strength SNR $\approx 19$, where the difference between SUPER-Net and Ensemble is not statistically significant ($p = 0.09$).

## Appendix C. Clinical Dataset

We include a clinical MRI dataset acquired at the University of Alabama at Birmingham (UAB), consisting of 627 FLAIR volumes from patients with grade II glioma (Fathallah-Shaykh et al., 2019). Each volume includes manual tumor annotations provided by an expert neuroradiologist. As with the other datasets in this study, we apply standard preprocessing (intensity normalization and removal of empty slices) and split the data into an 80/20 train–validation partition.

For this dataset, we use a U-Net backbone (details in Table 8) within the SUPER-Net framework. The model achieves a DSC of 86% on the validation set. Notably, it is also

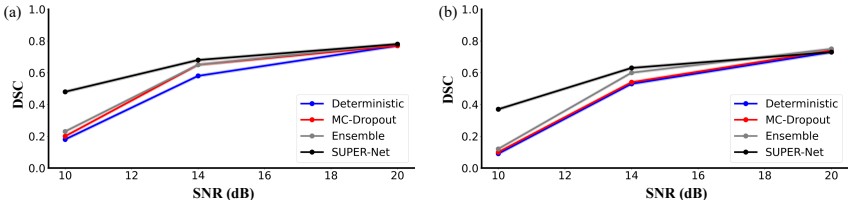

Figure 19: Mean DSCs are shown with shaded 95% confidence intervals for (a) Anterior and (b) Posterior Hippocampus for various levels of Speckle noise applied to the whole image of the Hippocampus test data.

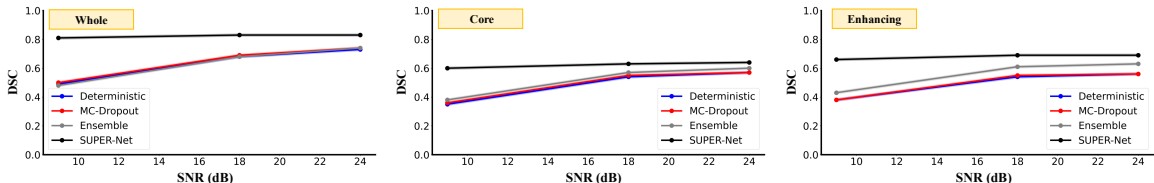

Figure 20: Mean DSCs are shown with shaded 95% confidence intervals for Gaussian noise applied to the BraTS data. Each subplot shows the Whole, Core, and Enhancing tumor structures.

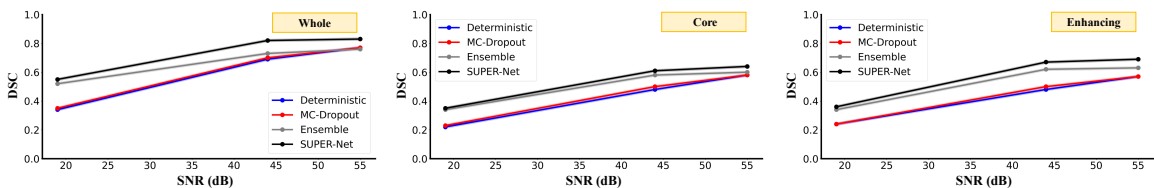

Figure 21: Mean DSCs are shown with shaded 95% confidence intervals for targeted adversarial attacks (source: label 3 → target: label 1) applied to the BraTS dataset. Each subplot shows the Whole, Core, and Enhancing tumor structures.

Table 8: Model architecture and training details for the clinical dataset.

| Encoder filters | Decoder filters | Epochs | Batch size | Optimizer | Learning Rate |
|---|---|---|---|---|---|
| 16, 32, 64, 128, 256 | 128, 64, 32, 16 | 100 | 10 | Adam | 0.001 |

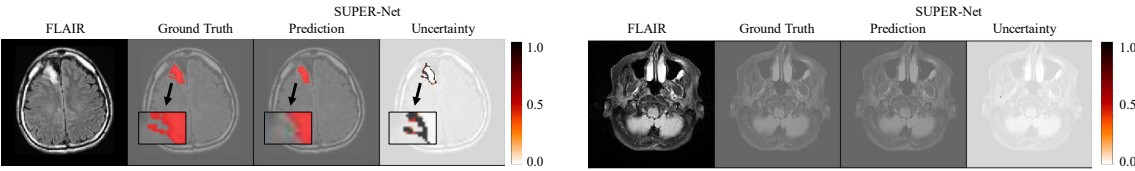

Figure 22: Two Sample scans from the clinical data. Both images show (left to right) the FLAIR input, the ground-truth segmentation, the SUPER U-Net prediction and the uncertainty map overlaid on the input scan. We zoom on regions incorrectly classified by the network and the corresponding regions in the uncertainty maps.

able to handle empty slices, i.e., scans without visible tumor, even though such cases were not included in the training set. Figure 22 illustrates two representative examples. In the first case, we highlight regions that are incorrectly segmented and compare them with the corresponding areas in the uncertainty map (computed from the predictive variance). As expected, the misclassified pixels exhibit high uncertainty. In the second example, we show an unseen empty scan. The model correctly predicts the absence of tumor, and, importantly, the associated uncertainty remains very low, indicating high confidence in its predictions.

