# OpenReview forum: "Testing the Trust: Verification and Validation of Bayesian Segmentation under Uncertainty"
_MIDL.io/2026/Validation_Papers — MIDL 2026 - Validation Papers Poster_

### Official Review · Reviewer_8F2w · 2026-01-02

**Confidence:** 4
**Preliminary Rating:** 4
**Final Rating:** 5

**Summary:**

This paper systematically evaluates the robustness and uncertainty quantification of state-of-the-art deterministic and Bayesian segmentation models in the context of medical image segmentation. It highlights the importance of trustworthiness in clinical AI applications, emphasizing that beyond accuracy, models must be reliable under real-world conditions, including noise and adversarial perturbations. Through a comprehensive verification and validation (V&V) framework, the paper demonstrates that Bayesian models, especially SUPER-Net, which propagate uncertainty during training, offer superior robustness and meaningful uncertainty estimates. These characteristics make them more suitable for deployment in clinical settings where model reliability and trust are crucial. The study covers three key datasets—Lung CT, Hippocampus MRI, and BraTS—and evaluates multiple noise types and adversarial attacks, showcasing the enhanced performance of uncertainty-aware models in the face of distributional shifts.

**Strengths:**

The paper excels in presenting a comprehensive and systematic evaluation of both deterministic and Bayesian models, providing critical insights into model behavior under noise and adversarial attacks. By emphasizing the need for rigorous verification and validation (V&V) in clinical AI, the paper contributes to improving the robustness and trustworthiness of AI systems. The use of multiple datasets, real-world noise types, and adversarial perturbations ensures that the findings are generalizable and relevant to practical applications. Furthermore, the emphasis on uncertainty propagation as a key feature for improving clinical decision-making is highly valuable. The SUPER-Net model, which integrates uncertainty propagation, demonstrates a clear advantage over deterministic models in terms of both robustness and interpretability, making it a strong candidate for clinical applications. The paper also addresses the critical issue of uncertainty quantification, which is often underexplored in medical AI systems.

**Weaknesses:**

While the paper is robust in its evaluation of uncertainty-aware models, it lacks a direct comparison of the clinical outcomes that could result from these models versus traditional methods in real-world scenarios. It would be beneficial to provide examples or case studies that illustrate how uncertainty estimates can actively aid clinicians in decision-making. Additionally, while the paper emphasizes the need for uncertainty-aware models, it could expand on the practical implications of deploying such models in clinical environments, including the challenges related to interpretability and the integration of these models into existing workflows. The reliance on synthetic noise and perturbations, while important for robustness testing, may not fully capture the complexities and variations of real-world clinical data. More discussion on how these models would perform on diverse patient populations and data from varied clinical sources would strengthen the paper’s findings.

**Detailed Comments:**

The paper could benefit from a section discussing the challenges of incorporating uncertainty-aware models into clinical practice, especially concerning clinician trust in AI systems.

The results would be more impactful with a clearer comparison between uncertainty-aware models and existing clinical segmentation tools, particularly in terms of diagnostic accuracy and workflow integration.

The discussion on adversarial attacks and distributional shifts could be expanded to include more real-world examples or specific clinical scenarios where these models would be applied.

**Justification Of Final Rating:**

The authors have made substantial progress in addressing the concerns raised in the initial review. Their detailed rebuttal provides concrete examples and case studies that illustrate how uncertainty propagation can aid in clinical decision-making, particularly in the context of brain tumor segmentation. By citing real-world scenarios, such as changes in scanner hardware or previously unseen tumor morphologies, the authors have effectively demonstrated the clinical relevance of uncertainty-aware models. The added clarity on how uncertainty maps serve as visual warning signals in clinical workflows strengthens the paper's practical impact.

The authors have also provided a comprehensive vision for integrating uncertainty-aware models into clinical workflows. Their approach of using uncertainty maps as decision-support tools aligns with current clinical practices, ensuring that the models are deployed as aids to clinicians rather than autonomous systems. This addresses the reviewer’s concerns about clinician trust and the acceptance of AI predictions. Furthermore, the additional section on clinical implications and deployment considerations (Section 3.4) adds significant value to the paper, providing a roadmap for the real-world implementation of these models.

While the paper could benefit from more extensive discussion on the performance of these models across diverse patient populations and clinical data sources, the authors have effectively demonstrated the robustness and potential of uncertainty-aware models in clinical applications. The rebuttal and revisions have significantly enhanced the manuscript’s relevance to practical deployment in medical AI, addressing the key concerns raised in the review.

Therefore, the paper now merits acceptance due to its clear contributions to improving the robustness and interpretability of AI models in clinical settings, with practical insights into model deployment and clinician acceptance.

**Justification Of The Preliminary Rating:**

The paper makes a significant contribution to medical AI by emphasizing the importance of uncertainty quantification and model robustness in clinical decision-making. The systematic evaluation of Bayesian and deterministic models across different datasets and perturbations demonstrates the value of uncertainty-aware models like SUPER-Net in clinical settings. However, the paper would benefit from a deeper exploration of the clinical implications and real-world deployment challenges. Additionally, while the uncertainty analysis is comprehensive, the clinical context in which these models will be used remains underexplored.

**Questions To Address In The Rebuttal:**

Can you provide specific examples or case studies where uncertainty propagation helped improve clinical decision-making?

How do you envision the deployment of uncertainty-aware models in real-world clinical workflows, especially concerning clinician training and acceptance of AI predictions?

---

### Official Review · Reviewer_TwQw · 2026-01-08

**Confidence:** 3
**Preliminary Rating:** 3
**Final Rating:** 4

**Summary:**

This paper addresses the problem of trustworthiness and robustness in medical image segmentation, with a particular focus on whether Bayesian and uncertainty-aware segmentation models genuinely provide more reliable behavior under realistic distributional shifts. Rather than proposing a new segmentation algorithm, the authors design a comprehensive verification and validation (V&V) framework to systematically evaluate deterministic, approximate Bayesian (MC-Dropout, Ensembles), and uncertainty-propagating models (SUPER-Net). Experiments span multiple public datasets, corruption types (noise and adversarial attacks), and clinically motivated metrics beyond Dice, including HD95 and over-/under-segmentation. The study demonstrates that models explicitly learning and propagating uncertainty tend to show improved robustness and more meaningful uncertainty signals under severe perturbations, supporting their suitability for safety-critical clinical deployment.

**Strengths:**

1. Multiple datasets, corruption types (Gaussian, Poisson, Speckle, Salt-and-Pepper, adversarial), and metrics are considered, providing a broad and rigorous validation.
2. The inclusion of HD95 and over-/under-segmentation rates, along with clear clinical motivation, goes beyond standard accuracy-centric evaluation.
3. The paper examines not only whether uncertainty is higher for incorrect predictions, but also whether uncertainty scales monotonically with corruption severity, which is important for trustworthiness.

**Weaknesses:**

1. Although trends are clear, the paper rarely reports statistical tests or confidence intervals (beyond cross-validation summaries) to support claims of superiority or robustness differences.
2. While positioned as a validation study, a substantial portion of the analysis emphasizes the advantages of SUPER-Net, which may limit the perceived neutrality of the comparison.
3. FGSM and PGD are standard, but the clinical plausibility of these perturbations is not fully discussed, which weakens the translational relevance of the adversarial results.

**Detailed Comments:**

1. Clarify whether all models were tuned with comparable effort (e.g., dropout rates, ensemble size) to ensure fairness in validation.
2. Add more explicit discussion on how the selected SNR ranges relate to real clinical acquisition scenarios.
3. Expand the explanation of how uncertainty thresholds are chosen when removing “uncertain pixels” in the ∆DSC analysis.

**Justification Of Final Rating:**

The authors have substantially strengthened the paper in response to the review, addressing several key concerns that previously limited confidence in the conclusions. In particular, the addition of statistical validation via bootstrap confidence intervals and paired significance testing meaningfully supports claims of robustness differences, and the new analysis of inference-time and memory complexity clarifies the practical trade-offs of the evaluated methods for clinical deployment. The rebuttal also convincingly argues that the observed robustness gains of uncertainty-propagating models are unlikely to be explained solely by increased capacity or generic regularization, given the shared backbone, modest parameter increase, and qualitatively different uncertainty behavior. While some limitations remain: (1) most notably the lack of an explicit ablation isolating uncertainty propagation effects, (2) limited sensitivity analysis for MC/ensemble hyperparameters, and (3) an incomplete discussion of the clinical realism of adversarial perturbations. However, the work now presents a careful, well-supported validation study with clear methodological contributions. Overall, the revisions sufficiently address the major concerns to warrant acceptance, despite remaining opportunities for further depth and refinement.

**Justification Of The Preliminary Rating:**

The experimental scope is broad, spanning multiple datasets, perturbation types, and clinically motivated metrics, and the methodology is clearly described. At the same time, several conclusions are based primarily on qualitative trends without statistical significance analysis, and the evaluation emphasizes a limited set of uncertainty formulations. Important validation aspects such as quantitative uncertainty calibration, computational cost, and the clinical realism of some perturbations are only partially addressed. Overall, the work demonstrates careful execution while leaving open questions regarding the strength and generality of its conclusions.

**Questions To Address In The Rebuttal:**

1. How do the authors ensure that the observed robustness gains of SUPER-Net are not primarily due to increased model capacity or regularization effects rather than uncertainty propagation itself?
2. Can the authors provide statistical significance testing to support claims of improved robustness or uncertainty quality?
3. Can the authors comment on inference-time cost and memory usage across methods, and how this affects clinical feasibility?
4. How sensitive are the conclusions to choices such as the number of MC samples, ensemble size, or noise parameterization?

---

### Official Review · Reviewer_JdxZ · 2026-01-09

**Confidence:** 4
**Preliminary Rating:** 3
**Final Rating:** 4

**Summary:**

This paper presents verification and validation for medical image segmentation, systematically evaluating deterministic and Bayesian models under noise, adversarial perturbations and distributional shifts. The results demonstrate that Bayesian models with producing uncertainty during forward (like SUPER-Net), provide superior robustness and more clinically meaningful uncertainty estimates beyond accuracy alone.

**Strengths:**

1. The paper clearly positions segmentation evaluation within a rigorous perspective, moving beyond conventional accuracy analysis.
2. Systematically experiments on multiple datasets, corruption types and diverse metrics.
3. The paper shows the insightful uncertainty analysis such as how and when the uncertainty correlated with the segmentation errors.

**Weaknesses:**

1. The experimental evaluation is restricted to a deterministic U-Net and approximate Bayesian variants. It limits the comparison against more recent segmentation paradigms, such as transformer-based models.
2. The paper does not provide detailed setting information about the attack configurations for the FGSM and PGD.

**Detailed Comments:**

It would be better to include comparisons with more recent state-of-the-art segmentation methods, beyond the deterministic U-Net and its Bayesian variants. In addition, the paper does not clearly specify whether the data splits are performed at the patient level or slice level.

**Justification Of Final Rating:**

My preliminary rating is based on the paper’s systematic verification and validation-oriented evaluation of segmentation robustness and uncertainty. The rating is moderated by the limited range of compared models, missing details on adversarial attack configurations. In the rebuttal, the authors have addressed most of the concerns, so I raise the final rating.

**Justification Of The Preliminary Rating:**

My preliminary rating is based on the paper’s systematic verification and validation-oriented evaluation of segmentation robustness and uncertainty. The rating is moderated by the limited range of compared models, missing details on adversarial attack configurations.

**Questions To Address In The Rebuttal:**

None

---

### Author Rebuttal · Authors · 2026-01-24

**Rebuttal:**

We are grateful for the careful reviews and for highlighting both the strengths and areas for improvement in our work. We believe that incorporating the reviewers’ suggestions has significantly strengthened the manuscript and clarified the scope, methodology, and contributions of our study.

We have incorporated the reviewer’s suggestions throughout the manuscript, marking all new additions in red to clearly indicate changes. Below, we provide official responses addressing each reviewer point, including clarifications and additional comments that were raised in the full reviews.

**Supporting Material:**

/attachment/e2fbbc274d2c3e333cfe7365c8be28085bfe19e2.pdf

---

### Meta-Review · Area_Chair_t5D9 · 2026-02-09

**Recommendation:** Accept (Poster)
**Confidence:** 4

**Metareview:**

This paper presents a systematic verification-and-validation framework for medical image segmentation, focusing on robustness and uncertainty behavior across noise, adversarial perturbations, and distributional shifts. All three reviewers recognize the breadth of the experimental design, the clinically motivated evaluation metrics, and the clear positioning of the work beyond accuracy-only analyses. The study is consistently viewed as careful, well executed, and relevant to trustworthy clinical AI.

In the rebuttal, the authors clarified the experimental protocol, added attack configurations, provided statistical analyses, discussed computational trade-offs, and expanded the clinical interpretation. Reviewers acknowledged these improvements, and all three raised their scores, with final ratings of 4, 4, and 5.

Overall, the paper makes a solid contribution as a systematic validation study. The methodology is sound, the experimental scope is broad, and the revisions adequately address the reviewers’ main concerns. While some limitations remain (e.g., scope of compared models and depth of sensitivity analyses), the consensus is that the work is meaningful and suitable for MIDL.

---

### Decision · Program_Chairs · 2026-02-14

Accept (Poster)